# Observation of entanglement transition of pseudo-random mixed states

Tong Liu [1,2], Shang Liu[3], Hekang Li [1], Hao Li[1], Kaixuan Huang[1,4], Zhongcheng Xiang[1,2,4,5,6,7], Xiaohui Song[1,2,4,5,6,7], Kai Xu [1,2,4,5,6,7] ✉, Dongning Zheng[1,2,4,5,6,7] ✉ & Heng Fan [1,2,4,5,6,7] ✉

Random quantum states serve as a powerful tool in various scientific fields, including quantum supremacy and black hole physics. It has been theoretically predicted that entanglement transitions may happen for different partitions of multipartite random quantum states; however, the experimental observation of these transitions is still absent. Here, we experimentally demonstrate the entanglement transitions witnessed by negativity on a fully connected superconducting processor. We apply parallel entangling operations, that significantly decrease the depth of the pseudo-random circuits, to generate pseudo-random pure states of up to 15 qubits. By quantum state tomography of the reduced density matrix of six qubits, we measure the negativity spectra. Then, by changing the sizes of the environment and subsystems, we observe the entanglement transitions that are directly identified by logarithmic entanglement negativities based on the negativity spectra. In addition, we characterize the randomness of our circuits by measuring the distance between the distribution of output bit-string probabilities and the Porter-Thomas distribution. Our results show that superconducting processors with all-to-all connectivity constitute a promising platform for generating random states and understanding the entanglement structure of multipartite quantum systems.

Random quantum states, sampled from Haar measure, have broad applications in quantum supremacy[1,2], quantum communication[3], quantum metrology[4], and fidelity benchmarking[5,6]. In addition, reduced density matrices of random states have also attracted extensive interest owing to the strong connection between quantum chaos and black hole physics[7–10]. Entanglement is a crucial property of quantum states[11–17], and it is expected that random quantum states may hold universal entanglement characteristics, which can be classified into different phases[18–22].

The classification of phases is based on the entanglement between two subsystems of a tripartite random state, with the remaining party regarded as an environment. The entanglement between subsystems is quantified by the negativity, which is an effectively computable measure of entanglement for quantum states, particularly for mixed states[23–27]. Negativity also plays a significant role in diagnosing finite-temperature topological order of toric code[28], entanglement properties of diffusive fermion model[29], replica symmetry breaking in holography[30] and finite-temperature phase transitions[31]. By modifying

[1]Institute of Physics, Chinese Academy of Sciences, Beijing 100190, China. [2]School of Physical Sciences, University of Chinese Academy of Sciences, Beijing 100190, China. [3]Kavli Institute for Theoretical Physics, University of California, Santa Barbara, CA 93106, USA. [4]Beijing Academy of Quantum Information Sciences, Beijing 100193, China. [5]Hefei National Laboratory, Hefei 230088, China. [6]CAS Center of Excellence for Topological Quantum Computation, University of Chinese Academy of Sciences, Beijing 100190, China. [7]Songshan Lake Materials Laboratory, Dongguan 523808 Guangdong, China. ✉e-mail: kaixu@iphy.ac.cn; dzheng@iphy.ac.cn; hfan@iphy.ac.cn

the sizes of subsystems and environment, a transition from positive partial transpose (PPT) states with a vanishing negativity to negative partial transpose (NPT) states with a non-zero negativity was numerically predicted in the large Hilbert space limit[19,20]. Recently, another phase transition between two types of NPT states was uncovered theoretically[22]. However, the experimental observation is still absent since the complexity of the generation of random quantum states grows exponentially with the system size[32], and the measurement of entanglement is experimentally challenging[33–42].

In this work, we report our experiments in probing entanglement negativity transition for pseudo-random quantum states using a fully connected superconducting quantum processor. A fully connected quantum processor may enhance the entangling power of shallow circuits compared with short-range connected processors and facilitate the realization of random quantum states. We apply the parallel entangling gates to realize the pseudo-random quantum circuits for generating pseudo-random states of 7–15 qubits, which can mimic specific statistical properties of random states[43–50]. The all-to-all architecture of our processor, which has been suggested to realize polynomial or exponential improvements in some quantum algorithms[51,52], helps to decrease the circuit depth and exposure to noise. Then, we utilize the quantum state tomography (QST) to directly obtain reduced density matrices of subsystems with six qubits[53–58] and calculate the negativity spectra. Other proposals to explore negativity require either multiple copies of target states[25,27] or global random unitary operators[59], which are both demanding for the present noisy intermediate-scale quantum devices. Our results represent the first experimental investigation of the entanglement negativity transition for random quantum states. It should be noted that many efforts have been made in studying negativity experimentally[60,61]. Our results are established on the reconstruction of reduced density matrices, which contain complete entanglement information in the mixed states.

## Results

Our processor contains 20 frequency-tunable transmon qubits and one central resonator. We use 15 qubits in the experiments and tune the frequencies of remaining qubits lower than 4 GHz. All qubits are capa-citively coupled to the central resonator, as shown in Fig. 1a, where qubits used in the experiments are labeled by $Q_j$ with $j \in \{1, 2, \ldots, 15\}$ and the central resonator is denoted as $\mathcal{R}$. Each qubit can be addressed by its separate XY line and Z line, which allows us to apply single-qubit or multi-qubit gates to specified qubits. Each layer of the pseudo-random circuit is composed of $N$ random single-qubit gates sampled from Haar measure on the SU(2) group and a global entangling gate $U$. By controlling the amplitudes and phases of Gaussian-enveloped microwave pulses transmitted by the XY lines, as shown in Fig. 1b, we can fulfill different rotation gates $R_\varphi(\theta)$ within a 15 ns duration $\tau_{rot}$, where $R_\varphi(\theta) = e^{-i(\cos\varphi\sigma^x + \sin\varphi\sigma^y)/2}$. In order to realize a random single-qubit gate, we decompose each single-qubit gate into two successive rotation gates $R_\varphi(\theta)$ of which rotation axes both lie in the $xy$ plane[62]. The global entangling gate $U$ acting on the $N$ qubits is defined as

$$U = \exp\left[-i\tau_{ent}\sum_{i<j}^{N} J_{ij}\left(\sigma_i^+ \sigma_j^- + \sigma_j^+ \sigma_i^-\right)\right], \tag{1}$$

where $\sum_{i<j}^{N} J_{ij}(\sigma_i^+ \sigma_j^- + \sigma_j^+ \sigma_i^-)$ is the effective Hamiltonian of selective $N$ qubits by equally detuning them from resonator $\mathcal{R}$ with the other qubits being far off-resonant[63,64]. $\sigma_j^+$ ($\sigma_j^-$) is the raising (lowering) operator for $Q_j$, $J_{ij}$ is the effective coupling strength between $Q_i$ and $Q_j$ (Fig. 1c), and $\tau_{ent}$ is the evolution time of about 40 ns. As the layer of circuits increases, the measure over pseudo-random circuits converges to the Haar measure exponentially though the rate of convergence decreases exponentially with the number of qubits[43,65].

By dividing all qubits into three parts, $A_1$, $A_2$ and $B$, which are comprised of $N_{A_1}$, $N_{A_2}$ and $N_B$ qubits, respectively, we regard the union of $A_1$ and $A_2$ as a system $A$ and $B$ as the environment of $A$. After applying a $d$-layer pseudo-random circuit to the initial state $|0\rangle^{\otimes N}$, we perform QST on the system qubits to estimate the reduced density matrix $\rho_A$. The tomography of states relies on measuring all system qubits in the eigenvectors of $\sigma_j^x$, $\sigma_j^y$, and $\sigma_j^z$. The measurement of $\sigma_j^z$ is direct by defining $\sigma_j^z \equiv |0_j\rangle\langle 0_j| - |1_j\rangle\langle 1_j|$. By inserting a $\pi/2$ rotation pulse X/2 (Y/2) before the readout pulse of $Q_j$, we can measure the state in the $\sigma_j^y$ ($\sigma_j^x$) basis. The whole pulse sequence for the QST including the state generation, the tomography operation, and the readout takes about

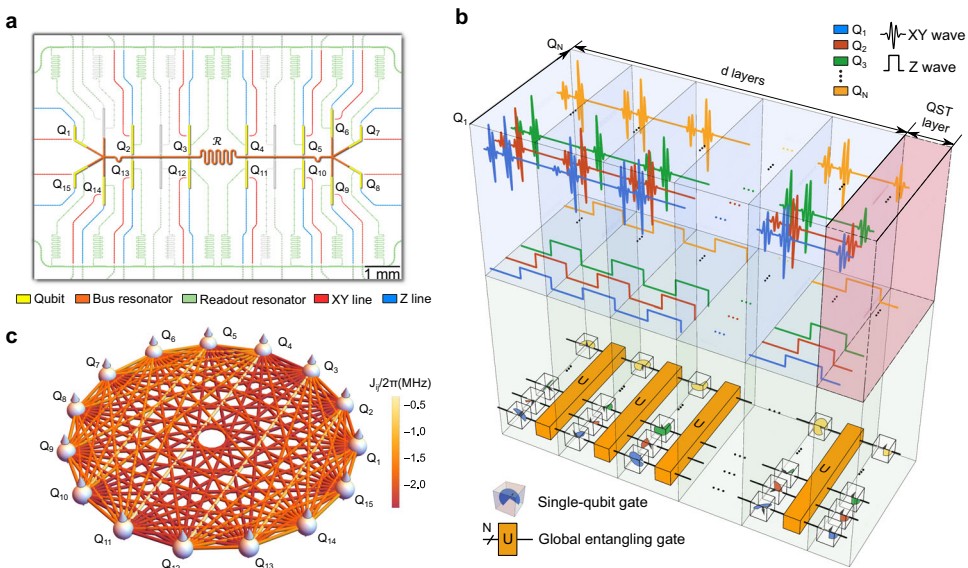

**Fig. 1 | Quantum simulator and experimental pulse sequences. a** False-color optical micrograph with highlighting various circuit elements. Qubits (yellow) are labeled from $Q_1$ to $Q_{15}$ and the central resonator (red) capacitively coupled to all qubits is labeled as $\mathcal{R}$. Each labeled qubit can be controlled by its XY line (red) and Z line (blue), and measured through the readout resonator (green). The device used here is the same one used in refs. 63, 69. **b** The pulse sequences of a pseudo-random circuit and its equivalent gate model. $U$ is a global entangling gate and each cubic box is a random single-qubit gate. **c** The schematic representation of the effective coupling graph of 15 qubits with an equal detuning $\Delta/2\pi \approx -360$ MHz from the central resonator.

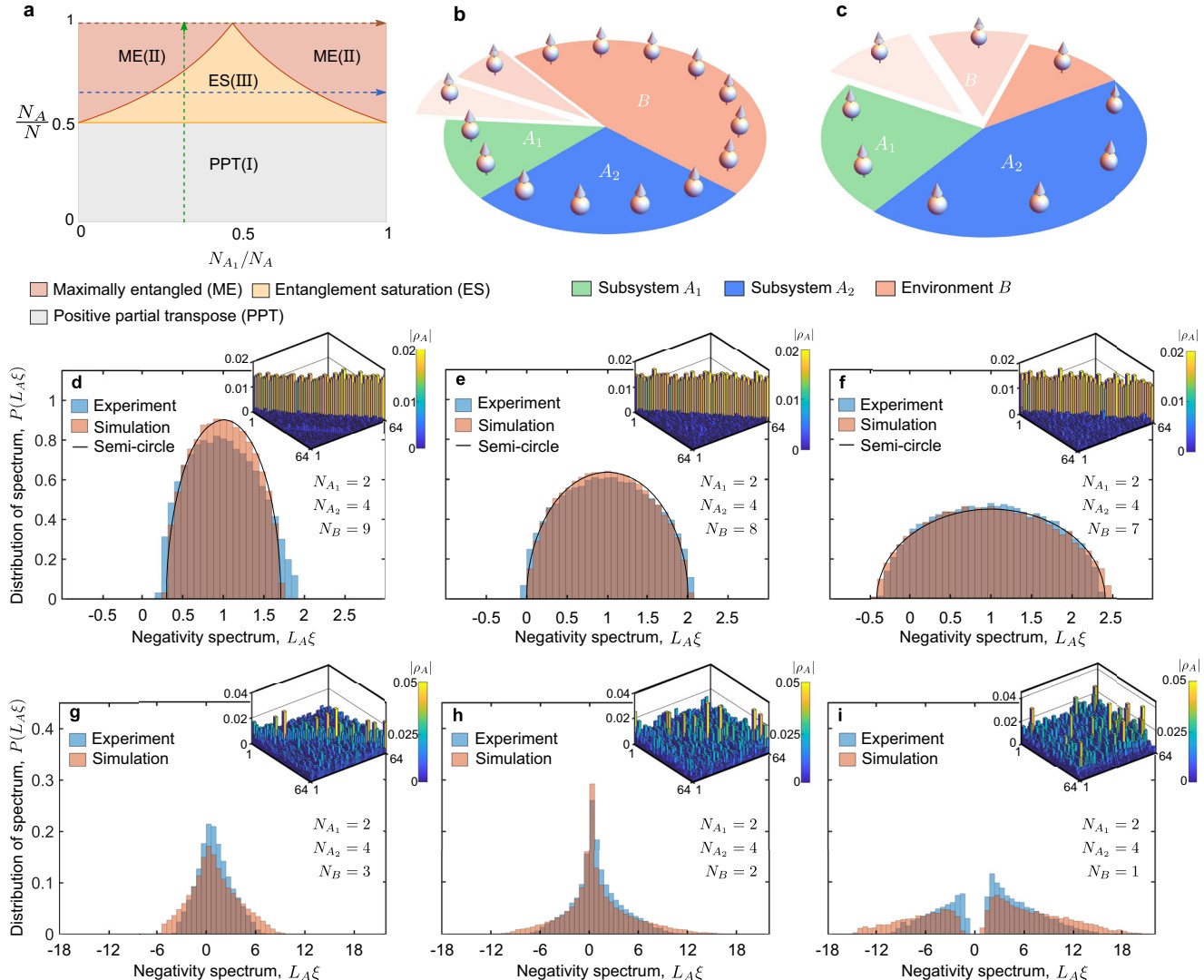

**Fig. 2 | Phase diagram and negativity spectra. a** Analytical phase diagram of reduced density matrix $\rho_A$ when $N \to \infty$. Green arrow indicates a path along which the negativity is plotted in Fig. 3a. Blue or brown arrows indicate paths along which the negativity is plotted in Fig. 3b. **b, c** Cartoons of subsystems and environment. Green, blue, and orange sectors represent subsystems $A_1$, $A_2$ and environment $B$, respectively. **d–f** Negativity spectra sampled from pseudo-random circuits where $N_{A_1} = 2$, $N_{A_2} = 4$ and $N_B = 9$, 8, or 7. **g–i** Negativity spectra sampled from pseudo-random circuits where $N_{A_1} = 2$, $N_{A2} = 4$ and $N_B = 3$, 2 or 1. One of the density matrices sampled from circuits with different sizes of environment is shown in the northeast corner of **d–i**.

two $\mu$s. With $3^{N_A}$ tomography operations and $2^{N_A}$ probabilities for each operation, we can reconstruct the state of system qubits (see Supplementary Note 5). Then we calculate the negativity $\mathcal{N}_{A_1:A_2}$ between $A_1$ and $A_2$ according to

$$\mathcal{N}_{A_1:A_2} = \frac{||\rho_A^{T_1}||_1 - 1}{2}, \quad (2)$$

where $||O||_1 = \text{Tr}(\sqrt{O^\dagger O})$ is the trace norm and $\rho_A^{T_1}$ represents the partial transpose of the density matrix of $A$ with respect to the subsystem $A_1$. $\mathcal{N}_{A_1:A_2}$ can also be written as the absolute value of sum of all negative eigenvalues of $\rho_A^{T_1}$. All eigenvalues of $\rho_A^{T_1}$ constitute the negativity spectrum. Another associated entanglement measure, logarithmic negativity, can be deduced from negativity by $\mathcal{E}_{A_1:A_2} = \log(2\mathcal{N}_{A_1:A_2} + 1)$. In the following context, the negativity is referred to as the logarithmic negativity.

The volume or size $L_i$ of part $i$ is defined as the dimension $2^{N_i}$ of the Hilbert space $\mathcal{H}_i$ where $i = A_1$, $A_2$ and $B$. $\rho_A$ is a PPT state if $L_A \equiv 2^{N_A} = 2^{N_{A_1} + N_{A_2}} > L_B/4$; otherwise, it is an NPT state[19,20,22]. NPT states can be furthermore classified into maximally entangled (ME) states and

entanglement saturation (ES) states via negativity[22]. The phase diagram of the reduced density matrix $\rho_A$ (when $N \to \infty$) shown in Fig. 2a, is divided into three phase regions PPT (I), ME (II), and ES (III), dependent on the ratio $N_{A_1}/N_A$ and $N_A/N$[22]. To characterize the transition from PPT to NPT, which occurs at $N_B = N_A + 2$[19,20,22], we fix the sizes of two subsystems as $N_{A_1} = 2$ and $N_{A_2} = 4$, and decrease the number of environment qubits $N_B$ from 9 to 7 by biasing the unused qubits far off-resonant, as shown in Fig. 2b. After drawing 20 instances of pseudo-random circuits with five layers, of which depth is enough to capture statistical features in the simulation (see Supplementary Note. 3), the negativity spectra of $\rho_A$ for different environment sizes are illustrated in Fig. 2d–f. The distribution of the negativity spectrum is in close agreement with the semi-circle law[18–22], i.e.,

$$P(\xi) = \frac{2L_A}{\pi a^2} \sqrt{a^2 - \left(\xi - \frac{1}{L_A}\right)^2}, |\xi - \frac{1}{L_A}| < a, \quad (3)$$

where $P(\xi)$ is the probability density of negativity spectrum and $a \equiv 2/\sqrt{L_A L_B}$ is the radius. Note that $N_{A_1}$, $N_{A_2}$ and $N_B$ are chosen to satisfy $L_B L_{A_2} \gg L_{A_1}$ to meet the prerequisite of semi-circle law[19,20,22]. When

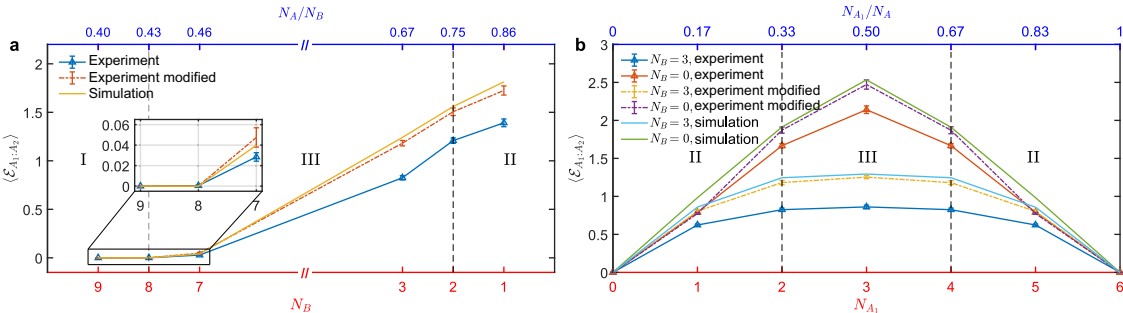

**Fig. 3 | Averaged logarithmic negativity $\langle \mathcal{E}_{A_1:A_2} \rangle$.** The error bars represent standard error of the mean over circuit instances. **a** Fix $N_{A_1} = 2$ and $N_{A_2} = 4$, then decrease $N_B$ from 9 to 1. **b** Fix $N_B = 3$ or 0 and increase $N_{A_1}$ from 0 to 6. When $N_B = 0$, the negativity approximately obeys volume law. In contrast, the negativity is saturated from $N_{A_1} = 2$ to 4 when $N_B = 3$.

$N_B = 9$, the negativity spectrum contains no negative values, which indicates that the system belongs to the PPT phase. Then we remove one environment qubit, sample new pseudo-random circuits, and apply them to the remaining qubits. Now the minimum of the negativity spectrum is close to zero, which shows an expected correspondence with the phase transition condition. Repeating the above procedure for seven environment qubits yields the distribution of the negativity spectrum which partly covers the negative domain, ensuring the existence of non-zero negativity between subsystems.

The second phase transition from ES to ME occurs at $|N_{A_1} - N_{A_2}| = N_B$. We still keep $N_{A_1} = 2$ and $N_{A_2} = 4$, and lower $N_B$ from 3 to 1 to detect the phase transition. The negativity spectra drawn from 20 instances of four-layer pseudo-random circuits are shown in Fig. 2g–i. In contrast with the negativity spectra obtained for $N_B \geq 7$, the negativity spectrum for $N_B = 3$ has a wider distribution and the center of the distribution is close to zero, as displayed in Fig. 2g. In Fig. 2h, we can observe a sharp peak located at zero emerging in the distribution of negativity spectrum with $N_B = 2$, which is diverging for $N \to \infty$ [22]. Next, we set $N_B = 1$ and show the distribution of negativity spectra in Fig. 2i where we exclude some eigenvalues in the vicinity of zero (see Supplementary Note. 4). The remaining eigenvalues are split into two disjoint parts. The distribution of each part can be approximated by the Marčenko-Pastur distribution[22].

Figure 3a incorporates the logarithmic negativities derived from the negativity spectra for different environment sizes. We average the negativities over all combinations of subsystem qubits for each sampled density matrix. We also show the negativities derived from the density matrices modified to alleviate decoherence errors in Fig. 3, where we use truncated eigenvalue decomposition such that the purity of the modified density matrix is close to the mean purity of density matrices sampled without decoherence errors (see Supplementary Note. 6). We start from the PPT (I) phase with zero negativity, enter the ES (III) phase at $N_B = 8$, and arrive at the ME (II) phase when $N_B = 2$. The related path in the phase diagram can be described by the vertical green line shown in Fig. 2a. To distinguish the ME and SE phases, we change the ratio between the two subsystems $A_1$ and $A_2$, but keep the environment invariant. Another benefit of QST is that we can compute the negativity between any two parts of the system without additional measurements. As a comparison to the aforementioned results, we also measure the density matrices of six system qubits without environment qubits. When $N_B = 0$, the negativity grows linearly with the number of $N_{A_1}$, which resembles a Page curve. However, the growth of negativity is depressed and saturated at $N_{A_1} = N_{A_2} = 3$ when considering three environment qubits. These results can be interpreted heuristically as follows: First, if there exists no environment $B$, system $A$ is totally entangled. Thus, the entanglement between two subsystems is proportional to the size of the minimal one. Second, if subsystem $A_1$ is larger than $A_2$ plus $B$, $A_2$ and $B$ will be entangled entirely to $A_1$. Then we can deduce that there are $N_{A_2}$ pairs of entangled qubits in system $A$. Since the number of maximally entangled pairs between $A_1$ and $A_2$ is $N_{A_2}$, we call this phase the maximally entangled phase. Third, if $A_1$ and $A_2$ are comparable in size and $(N_{A_1} + N_{A_2}) > N_B$, environment $B$ will be entangled with $A_1$ and $A_2$ in a way where $A_1$ and $A_2$ have the same number of remaining qubits to entangle with each other. Hence, the entanglement between $A_1$ and $A_2$ is roughly $(N_A - N_B)/2$, and we call this phase the entanglement saturation phase. Finally, these results can be recapped by the following formula[22]

$$\langle \mathcal{E}_{A_1:A_2} \rangle \approx \begin{cases} 0, & N_A < N_B, \\ \frac{1}{2}(N_A - N_B) + c, & N_{A_s} < \frac{N}{2}, s = 1,2 \text{ and } N_A > N_B, \\ \min(N_{A1}, N_{A2}), & \text{otherwise}, \end{cases} \quad (4)$$

where $c = \log(8/3\pi)$.

Another distinctive aspect of random circuits is that the output bit-string probabilities $p(x) \equiv |\langle x|\psi \rangle|^2$ approaches the Porter-Thomas (PT) distribution, i.e., $\Pr(Lp) = e^{-Lp}$, with increasing depth, where $|\psi\rangle$ is the output state of a circuit and $x \in \{0, 1\}^N$ [1,2,66-68]. Figure 4a illustrates three histograms of the full output bit-string probabilities collected from 300 pseudo-random circuit instances for nine qubits with layers $d = 2$, 3 and 4, where small probabilities ($<1/L$) show more often compared to the large probabilities ($>4/L$). The dark solid line in Fig. 4a represents the PT distribution. It is clear that the distribution from three-layer circuits is closest to the PT distribution. To quantify the distance between the measured distribution and the PT distribution over layers, we use the Kullback-Leibler divergence, defined as $D_{KL} = S(P_{meas}, P_{PT}) - S(P_{meas})$ where $S(P_{meas}, P_{PT})$ is the cross entropy between the measured distribution $P_{meas}$ and the PT distribution $P_{PT}$, and $S(P_{meas})$ is the self-entropy of the measured distribution[67,69]. $D_{KL}(\geq 0)$ is zero if and only if two distributions are identical. As shown in Fig. 4b, $D_{KL}$ reaches the minimum after three layers, which verifies the observation. Then $D_{KL}$ increases over layers attributed to the decoherence errors[67,69]. Although the output of three-layer circuits for nine qubits is closest to the PT distribution in experiments, we observe that three-layer circuits are not deep enough for the negativity spectra to converge (see Supplementary Note. 3), suggesting the states to be not random enough, hence we implement deeper circuits in the negativity experiments. See also Supplementary Note. 9 where we discuss the effect of decoherence on bitstring probability distributions.

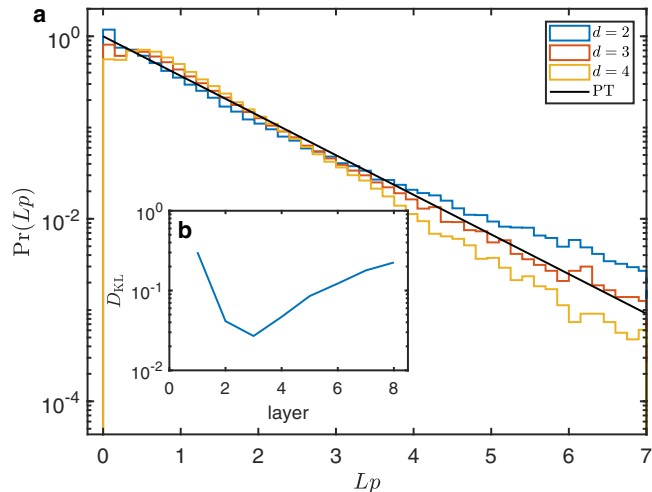

**Fig. 4 | Histograms of output bit-string probabilities sampling from pseudo-random circuits for nine qubits and KL divergence. a** Three histograms are sampled from two, three and four layers of circuits. Dark solid line represents the PT distribution. **b** The KL divergence between the sampling distribution and the PT distribution over layers.

## Discussion

We investigate the negativity spectrum of pseudo-random mixed states through QST on a fully-connected quantum processor. On account of the flexibility of tuning qubits, we can observe two phase transitions, from the PPT phase to the ES phase and from the ES phase to the ME phase, by biasing specified qubits far off-resonant. From the PPT phase to the ES phase, the distributions of negativity spectra can be well described by the semi-circle law[19,20,22]. From the ES phase to the ME phase, the distributions of negativity spectra in experiments are more concentrated at zero than in simulations due to decoherence errors. We also check the randomness of our circuits over different layers by comparing the distribution of probabilities of output bit-string with the Porter-Thomas distribution. The KL divergence between two distributions over layers decreases from 1 to 3 and increases from 3 to 8. Decoherence errors are the main obstructions in precisely measuring the negativity in experiments, which mix a random density matrix with a maximally mixed state and diminish the entanglement[1,2]. However, major features of the negativity spectrum of different phases can still be captured by our processor. Based on the pseudo-random circuit approach proposed in this work, our processor can also be a promising platform to study the measurement-induced phase transitions in all-to-all circuits by inserting local measurements[70,71], which can be realized after suppressing decoherence errors of the processor in the future.

## Methods
### Experimental device

As shown in Fig. 1a, the device consists of 20 transmon qubits coupled to a central resonator bus $\mathcal{R}$. In our experiment, we choose 15 qubits of them labeled by $Q_j$ with $j \in \{1,2,...,15\}$, which can be individually controlled by their XY lines and Z lines. The anharmonicities of qubits $\eta_j$ all lie within the range $-260$ MHz $< \eta_j/2\pi <$ $-240$ MHz. After arranging the idle frequency $\omega_j$ of each qubit by its Z line, we can decrease the crosstalk of XY lines between different qubits when performing single rotation gates such as X gates to all qubits simultaneously. In Supplementary Table 1, we list more details of 15 qubits used in the experiment. The frequency of central resonator $\mathcal{R}$ is fixed at $\omega_r/2\pi = 5.51$ GHz. The full Hamiltonian of the device can be written as ($\hbar = 1$)

$$H = \omega_r a_r^\dagger a_r + \sum_{j=1}^{15}\left[\omega_j |1_j\rangle\langle 1_j| + g_j\left(\sigma_j^- a_r + a_r^\dagger \sigma_j^+\right)\right] + \sum_{i<j}^{15} J_{ij}^c\left(\sigma_i^+ \sigma_j^- + \sigma_j^+ \sigma_i^-\right), \quad (5)$$

where $a_r$ ($a_r^\dagger$) denotes the annihilation (creation) operator of $\mathcal{R}$, $\sigma_j^-$ ($\sigma_j^+$) denotes the lowering (raising) operator of qubit $Q_j$, $g_j$ represents the coupling strength between qubit $Q_j$ and resonator $\mathcal{R}$, and $J_{ij}^c$ represents the small direct coupling between $Q_i$ and $Q_j$. By equally detuning the frequencies of all qubits from that of $\mathcal{R}$ by $\Delta$ and eliminating the resonator mode, we can realize the interactions between all pairs of qubits through the resonator. Now the Hamiltonian can be written as

$$H = \sum_{i<j}^{15} J_{ij}\left(\sigma_i^+ \sigma_j^- + \sigma_j^+ \sigma_i^-\right), \quad (6)$$

where $J_{ij} = J_{ij}^c + g_i g_j/\Delta$. The effective coupling strength $J_{ij}$, as shown in Supplementary Fig. 1, can be determined in the experiment through the single photon swapping process between $Q_i$ and $Q_j$.

### Sample from the Haar measure on SU(2)

As shown in the main text, we decompose an arbitrary SU(2) operator into two rotations of which rotation axes both lie in the $xy$ plane, so that the SU(2) operator can be described by the following parameters:
1. $\varphi$, the angle between the rotation axe of the first rotation and $x$ axis;
2. $\theta$, the rotation angle of the first rotation;
3. $\phi$, the angle between the rotation axis of the second rotation and $x$ axis.

The rotation angle of the second rotation is always $\pi$. Then the total operation can be written as

$$\begin{aligned} R_\phi(\pi)R_\varphi(\theta) = &- \sin\frac{\theta}{2}\cos(\varphi - \phi) - i\cos\phi\cos\frac{\theta}{2}\sigma^x \\ &- i\sin\phi\cos\frac{\theta}{2}\sigma^y - i\sin\frac{\theta}{2}\sin(\varphi - \phi)\sigma^z. \end{aligned} \quad (7)$$

Since any SU(2) operator sampled from Haar measure can be written as[32,72]

$$\begin{aligned} \begin{pmatrix} \cos\alpha\, e^{i\beta} & \sin\alpha\, e^{i\gamma} \\ -\sin\alpha\, e^{-i\gamma} & \cos\alpha\, e^{-i\beta} \end{pmatrix} = &\cos\alpha\cos\beta + i\sin\alpha\sin\gamma\,\sigma^x \\ &+ i\sin\alpha\cos\gamma\,\sigma^y + i\cos\alpha\sin\beta\,\sigma^z, \end{aligned} \quad (8)$$

where $\alpha$, $\beta$ and $\gamma$ are taken from the intervals

$$0 \leq \alpha \leq \frac{\pi}{2}, 0 \leq \beta < 2\pi, 0 \leq \gamma < 2\pi, \quad (9)$$

we obtain

$$\theta = \pi - 2\alpha, \quad \phi = \frac{\pi}{2} - \gamma, \quad \varphi = \phi + \beta. \quad (10)$$

For each random single-qubit gate in experiments, we firstly draw $\beta$ and $\gamma$ uniformly from the intervals in Eq. (9), then we draw another parameter $\xi$ from [0,1] uniformly and take the angle $\alpha$ as $\arcsin(\sqrt{\xi})$ [32,72]. Then we can obtain the experiment parameters $\theta$, $\phi$ and $\varphi$ by substituting $\alpha$, $\beta$ and $\gamma$ into Eq. (10). Besides, virtual $R_z$ gates are applied to each qubit after each frequency switch between the idle frequency $\omega_j$ and the entanglement frequency $\omega_{ent}$.

## Data availability
The datasets generated in this study have been deposited in the Zenodo repository, https://doi.org/10.5281/zenodo.7714334, and are available from the corresponding author H.F. upon request.

## Code availability
The codes for numerical simulation of pseudo-random circuits and data analysis are available from the corresponding author H.F. upon request.

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

## Acknowledgements

We thank J. Cui and Y.-R. Zhang for the fruitful discussions. This work was supported by National Natural Science Foundation of China (Grants Nos. T2121001, 92265207, 11904393, 92065114, 11875220, and 12047502), Innovation Program for Quantum Science and Technology (Grant No. 2-6), Strategic Priority Research Program of Chinese Academy of Sciences (Grant No. XDB28000000), Beijing Natural Science Foundation (Grant No. Z200009), and Scientific Instrument Developing Project of Chinese Academy of Sciences (Grant No. YJKYYQ20200041). S.L. is supported by Gordon and Betty Moore Foundation under Grant No. GBMF8690 and the National Science Foundation under Grant No. NSF PHY-1748958.

## Author contributions

K.X., D.Z., and H.F. supervised the research. T.L., S.L., K.X., and H.F. designed the experiment. H.L., Z.X., and X.S. fabricated the device. T.L., H.L., and K.H. performed the experiment. T.L. did numerical simulations. T.L., S.L., K.X., and H.F. analyzed the results. T.L. and H.F. wrote the paper. All authors contributed to the experimental setup, discussions of the results, and development of the manuscript.

## Competing interests

The authors declare no competing interests.
