## [Peer Review File · Nature Communications]

Observation of entanglement transition of pseudo-random mixed statesREVIEWER COMMENTS

Reviewer #1 (Remarks to the Author):

The manuscript by Liu et al. reports an experiment where a superconducting quantum circuit was used to generate pseudo-random mixed states and to observe phase transitions in their entanglement negativity spectra. For efficient sampling of random quantum states the authors employ a circuit with all-to-all connectivity between the 15 qubits used in experiment, which is mediated by a central bus resonator. The authors calculate negativity spectra from the density matrices of six so-called system qubits, which is measured in experiment by quantum state tomography. The manuscript closely follows a scheme theoretically introduced in Ref. 22 – a generalized theory for entanglement spectra of mixed states in a quantum system that couples to an environment. In the last part of their manuscript, the authors demonstrate the “randomness” of their sampled outputs by observing a Porter-Thomas distribution of the measured output probabilities.

As the sampling of random quantum states has been experimentally demonstrated several times before (also with larger quantum circuits and by using different randomization schemes), I view the last part of the paper (Fig. 4) mostly as a necessary prerequisite for the negativity experiment. While the sampling of random quantum states is not novel itself (apart from a possibly new gate sequence for the sampling procedure), the negativity experiment is however novel to my knowledge. The manuscript is written concisely, with the relevant theoretical background information introduced. The experimental results show good agreement with theoretical expectations and numerical simulations. The topic of the manuscript is timely and interesting to a general readership. In my opinion, the manuscript can therefore be a good fit for Nature Communications. Before I can give a final recommendation I would like to ask the authors to clarify several points as detailed in the following. The superconducting device used in this manuscript looks very similar to the one used in Ref. 62, even the frequency of the bus resonator is identical to the one in Ref. 62. Have the authors used the same chip as the one used in Ref. 62? I think a reference to Ref. 62 would be appropriate around Fig. 1. The authors mention in the text and in their abstract that their chip contains 20 transmon qubits. I recommend that the authors clearly state in their text that they use only 15 for their experiment, which is implied in the second sentence of the Results section. A comment on the frequency location of the five remaining qubits during the experiment would help to clarify (already in the first paragraph of Results). For the abstract, I suggest to only mention that 15 qubits are used in the experiment, since the third sentence may imply that all 20 qubits have been utilized.

According to Fig. 1c, the coupling strength between qubits is on the order of 0.5MHz to 2MHz. Can the authors provide some intuition as to how effective the 40ns free evolution per layer is to spread entanglement within the qubit lattice?

The authors mention that the rotation angles for the single-qubit rotations are “uniformly sampled” from certain intervals. I would appreciate some more details on this procedure. Are the rotation angles themselves random? If not, can the authors discuss how this may impact the randomness of the measurement outcomes?

In addition, I encourage the authors to discuss infidelities in the single-qubit rotation gates and variations in the qubit-qubit coupling strengths. Is the gate fidelity known or calibrated? I can imagine that certain infidelities may not harm the randomization process, but certain others which may be the result of unwanted correlations may influence the experimental outcome. Are interactions between qubits turned off during the single-qubit gates? Similarly, I wonder how differences or changes in qubit-qubit coupling strengths impact this experiment. Have the authors selected always the same six qubits as system qubits? Since the coupling map in Fig. 1c shows non-identical couplings between different pairs of qubits it might be a good sanity check to choose different qubits as the system qubits in successive experimental runs in order to support that variations in coupling strengths or gate infidelities do not impact the measurement outcome.

In Fig. 2a the authors show the phase diagram for infinitely large N , and it is my understanding that it was derived in Ref. 22 for the large-system limit. Can the authors comment on the validity of this phase diagram for finite system (and environment) sizes, as used in their experiment?

The authors mention that they used five layers for their experiments in Fig. 2 and Fig. 3. However, in Fig. 4 they show that the experimentally extracted probability distributions are closest to the expected Porter-Thomas distribution for three layers. Can the authors please elaborate on this discrepancy?

This suggests that the underlying probability distributions used for the negativity results do not follow a Porter-Thomas distribution. Logically, I would rather check that the probability distribution follows Porter-Thomas first (and find the optimal number of layers based on that) and then use this optimal layer number for the negativity experiments (basically show Fig. 4 first, followed by the negativity results). Can the authors comment why they did not adopt this strategy?

The authors do not mention the anharmonicity of their qubits nor the effect of the second excited levels of their transmon qubits. Did the authors check the higher-level excitations or can they motivate that there is no adverse effect for the probability distribution measurements, density matrix extraction, or their experimental scheme in general? Since the experimental realization of quantum state tomography in a six-qubit system is rather expensive in terms of number of measurements I would appreciate if the authors could provide some details on their experimental methods. I was not able to find the Supplementary Notes, where I assume the authors have made more explanations on the experimental details, but in any case I would appreciate it if the authors could add a few additional explanatory sentences on their experimental procedure in the main text or in the Methods section. This also applies to their correction scheme of data points, which deserves at least a short explanation in the main text.

In Fig. 3a, why is the negativity for $N_B=7$ so small, when according to the text we would expect non-zero negativity in the NPT region? Might a logarithmic scaling of the vertical axis in Fig. 3a help to better visualize these (small) values?

The authors explain how they divide their circuit into three parts A1, A2, and B in the last paragraph of page 3, but already use this notation in the context of negativity.

Fig 1a is missing a scale bar. In Fig. 1c the authors should invert their color bar, such that highest coupling strengths are not white.

Reviewer #2 (Remarks to the Author):

The work titled *Observation of entanglement transition of pseudo-random mixed states* by Tong Liu and coworkers have studied the Haar measure random states generated using superconducting processor. They have demonstrated the entanglement transition as predicted by works published in Refs. [19,20,21,22]. This is a very important experimental demonstration of its kind. The experimental results agrees very well the theoretical once considering the error introducing factors like decoherence. This system will also act as a testbed for testing other related results. The paper is well written with all key results explained very well. The figures are comprehensible. Key papers are included in the references. Despite these positives, I think authors need to implement following suggestions:

1. The authors should specify the total number of realizations i.e. total number of states generated (experiments and simulations) to get the plots of figures 2, 3 and 4. Also the time taken to generated one full state and its QST.
2. Some important references on QST can be added since it is one of the main tool in this work.
3. Authors can add the following relevant reference in the introduction: *Experimental Realization of a Measurement-Induced Entanglement Phase Transition on a Superconducting Quantum Processor* by Jin Ming Koh, Shi-Ning Sun, Mario Motta, and Austin J. Minnich (<https://arxiv.org/abs/2203.04338>).

Finally, I feel that the authors have experimentally demonstrated an important theoretical result on the entanglement transition. This work deserves the publication in *Nature Communications* and I am happy to recommend this work for publication.

Reviewer #3 (Remarks to the Author):

This work is a very nice experimental study of transitions in the structure of entanglement in quantum states generated by pseudorandom circuits. The novel experimental aspect of this study is the use of nonlocal connectivity, which allows for rapid scrambling. The reduced density matrices are characterized by standard tomographic techniques (and hence so is the negativity). The bit-string probabilities are also computed and found to agree with expectations for random circuits.

This work performs a nontrivial experimental test of an interesting theory prediction and as such I recommend its publication. The paper is clearly written and I have no substantive criticisms.

Responses to Reviewers

In what follows we first quote the reviewers' comments in black color and then provide our response after each point in blue.

Responses to the Reviewer #1

- **Remark 1 of Reviewer #1:** “The manuscript by Liu et al. reports an experiment where a superconducting quantum circuit was used to generate pseudo-random mixed states and to observe phase transitions in their entanglement negativity spectra. For efficient sampling of random quantum states the authors employ a circuit with all-to-all connectivity between the 15 qubits used in experiment, which is mediated by a central bus resonator. The authors calculate negativity spectra from the density matrices of six so-called system qubits, which is measured in experiment by quantum state tomography. The manuscript closely follows a scheme theoretically introduced in Ref. 22 – a generalized theory for entanglement spectra of mixed states in a quantum system that couples to an environment. In the last part of their manuscript, the authors demonstrate the “randomness” of their sampled outputs by observing a Porter-Thomas distribution of the measured output probabilities.

As the sampling of random quantum states has been experimentally demonstrated several times before (also with larger quantum circuits and by using different randomization schemes), I view the last part of the paper (Fig. 4) mostly as a necessary prerequisite for the negativity experiment. While the sampling of random quantum states is not novel itself (apart from a possibly new gate sequence for the sampling procedure), the negativity experiment is however novel to my knowledge. The manuscript is written concisely, with the relevant theoretical background information introduced. The experimental results show good agreement with theoretical expectations and numerical simulations. The topic of the manuscript is timely and interesting to a general readership. In my opinion, the manuscript can therefore be a good fit for Nature Communications. Before I can give a final recommendation I would like to ask the authors to clarify several points as detailed in the following.”

- **Our response:** We would like to express our sincere gratitude to the Reviewer for reviewing our paper, accurate summary of our work and constructive suggestions. We also particularly appreciate Reviewer's positive evaluation of our work that “The topic of the manuscript is timely and interesting to a general readership. In my opinion, the manuscript can therefore be a good fit for Nature Communications”. Based on the report, we have clarified several crucial points and improved the presentation.
- **Remark 2 of Reviewer #1:** “The superconducting device used in this manuscript looks very similar to the one used in Ref. 62, even the frequency of the bus resonator is identical to the one in Ref. 62. Have the authors used the same chip as the one used in Ref. 62? I think a reference to Ref. 62 would be appropriate around Fig. 1. The authors mention in the text and in their abstract that their chip contains 20 transmon

qubits. I recommend that the authors clearly state in their text that they use only 15 for their experiment, which is implied in the second sentence of the Results section. A comment on the frequency location of the five remaining qubits during the experiment would help to clarify (already in the first paragraph of Results). For the abstract, I suggest to only mention that 15 qubits are used in the experiment, since the third sentence may imply that all 20 qubits have been utilized.”

- **Our response:** We thank the Reviewer for raising this question. Here, Ref. 62 denotes reference “K.-M. Li, H. Dong, C. Song, and H. Wang, Approaching the chaotic regime with a fully connected superconducting quantum processor, Phys. Rev. A 100, 062302 (2019)”, and Ref. 57 denotes reference “C. Song, K. Xu, H. Li, Y.-R. Zhang, X. Zhang, W. Liu, Q. Guo, Z. Wang, W. Ren, J. Hao, H. Feng, H. Fan, D. Zheng, D.-W. Wang, H. Wang, and S.-Y. Zhu, Generation of multicomponent atomic Schrödinger cat states of up to 20 qubits, Science 365, 574 (2019)”. Yes, we have used the same chip as the one used in Ref. 62, and it is also the same chip as the one used in Ref. 57. We thank the valuable suggestion of Reviewer and add references to Ref. 62 and Ref. 57 in the caption of Fig. 1a. We also thank the suggestion to clearly clarify the number of qubits used in the experiments. We add a statement that we only use 15 qubits in the experiments and tune the frequencies of the five remaining qubits lower than 4 GHz in the first paragraph of Results Section. We also deleted “20-qubit” in the third sentence of abstract to avoid misunderstanding.
- **Remark 3 of Reviewer #1:** “ According to Fig. 1c, the coupling strength between qubits is on the order of 0.5MHz to 2MHz. Can the authors provide some intuition as to how effective the 40ns free evolution per layer is to spread entanglement within the qubit lattice?”
- **Our response:** We thank the Reviewer for noting this point. We first consider a simplest situation with only two qubits to get some intuition. After a free evolution with time t , we can obtain a gate

$$U = \exp[-iJ\tau(\sigma_1^+\sigma_2^- + \sigma_1^-\sigma_2^+)] = \begin{pmatrix} 1 & 0 & 0 & 0 \\ 0 & \cos Jt & -i \sin Jt & 0 \\ 0 & -i \sin Jt & \cos Jt & 0 \\ 0 & 0 & 0 & 1 \end{pmatrix}, \quad (1)$$

with a coupling strength J . If $Jt = \pi/4$, we can realize a \sqrt{i} SWAP-like gate which can generate a Bell state with a maximal entanglement by acting on an initial state $|01\rangle$. Then we consider a general situation with N qubits. It has been shown that multicomponent atomic Schrödinger cat states can be realized on a fully connected processor in Ref. 57 (Song 2019). After a free evolution with time $\pi/(mJ)$, where m is an integer no less than 2, the state will evolve to a superposition of m -component atomic cat states.

In our experiments, the average coupling strength between any two qubits among 15 used qubits is $\bar{J}/2\pi = 1.82$ MHz. To implement a \sqrt{i} SWAP-like gate, the evolution time

should be 68 ns. As shown in Supplementary Note 3, we found that similar statistical results can be achieved by decreasing the evolution time to 40 ns, which will save more than 100 ns after a four-layer circuit.

- **Remark 4 of Reviewer #1:** “ The authors mention that the rotation angles for the single-qubit rotations are “uniformly sampled” from certain intervals. I would appreciate some more details on this procedure. Are the rotation angles themselves random? If not, can the authors discuss how this may impact the randomness of the measurement outcomes? ”

Figure 1: Generation of a single-qubit gate in two steps.

- **Our response:** We thank the Reviewer for raising this important question. Yes, the rotation angles themselves are random and the rotation axes are also random. Our goal is to generate random single-qubit rotations from the Haar measure on SU(2) group. Any single-qubit rotation can be written as $e^{-i\alpha\hat{n}\cdot\sigma}$ where $\hat{n} = (\sin\beta\cos\gamma, \sin\beta\sin\gamma, \cos\beta)$ is the rotation axis and α is the rotation theta. To achieve that in our system, we decompose each single-qubit operation into two $R_\phi(\theta)$ gates of which rotation axis both lie in the $x - y$ plane as shown in Fig. 1. The rotation axis and angle of the first gate are both random. The rotation axis of the second gate is also random but the rotation angle is fixed as π . Thus, we have three freedoms to realize any single-qubit rotation. We can tune the rotation axis and angle by changing the phase and amplitude of Gaussian enveloped microwave pulse, respectively. Hence, the final rotation is

$$R_\phi(\pi)R_\phi(\theta) = -\sin\frac{\theta}{2}\cos(\varphi-\phi) - i\cos\phi\cos\frac{\theta}{2}\sigma^x - i\sin\phi\cos\frac{\theta}{2}\sigma^y - i\sin\frac{\theta}{2}\sin(\varphi-\phi)\sigma^z. \quad (2)$$

From Ref. 32 and Ref. 72, we know that any SU(2) operator can be written as

$$\begin{pmatrix} \cos\alpha e^{i\beta} & \sin\alpha e^{i\gamma} \\ -\sin\alpha e^{-i\gamma} & \cos\alpha e^{-i\beta} \end{pmatrix} = \cos\alpha\cos\beta + i\sin\alpha\sin\gamma\sigma^x + i\sin\alpha\cos\gamma\sigma^y + i\cos\alpha\sin\beta\sigma^z, \quad (3)$$

where α , β and γ are taken from the intervals

$$0 \leq \alpha \leq \pi/2, \quad 0 \leq \beta < 2\pi \quad \text{and} \quad 0 \leq \gamma < 2\pi. \quad (4)$$

Comparing Eq. (2) with Eq. (3) yields

$$\theta = \pi - 2\alpha, \quad \phi = \frac{\pi}{2} - \gamma, \quad \varphi = \phi + \beta. \quad (5)$$

For each random single qubit in experiments, we first draw β and γ uniformly from the intervals in Eq. (4), then we draw another parameter ξ from $[0, 1)$ uniformly and take the angle α as $\arcsin(\sqrt{\xi})$, according to the Haar measure on $SU(2)$ given in Ref. 32 and Ref. 72 of our revised manuscript

$$P(dU) \propto d(\sin \alpha)^2 d\beta d\gamma. \quad (6)$$

Then we can obtain the experiment parameters θ , ϕ and φ by substituting α , β and γ into Eq. (5).

In the Methods Section, we add more details on how to generate a single-qubit gate in our experiment and how to sample experiment parameters α , β and γ as stated in this response.

	Q ₁	Q ₂	Q ₃	Q ₄	Q ₅	Q ₆	Q ₇	Q ₈	Q ₉	Q ₁₀	Q ₁₁	Q ₁₂	Q ₁₃	Q ₁₄	Q ₁₅
Fidelity	99.15%	99.98%	99.68%	99.97%	99.99%	99.98%	99.89%	99.85%	99.87%	99.86%	99.98%	99.92%	99.59%	99.29%	99.96%

Figure 2: State fidelities of all qubits after a layer of random XY gate.

Figure 3: Frequency arrangement of 15 qubits when applying single-qubit gates.

- **Remark 5 of Reviewer #1:** “ In addition, I encourage the authors to discuss infidelities in the single-qubit rotation gates and variations in the qubit-qubit coupling strengths. Is the gate fidelity known or calibrated? I can imagine that certain infidelities may not harm the randomization process, but certain others which may be the result of unwanted correlations may influence the experimental outcome. Are interactions between qubits turned off during the single-qubit gates? Similarly, I wonder how differences or changes in qubit-qubit coupling strengths impact this experiment. Have the authors selected always the same six qubits as system qubits? Since the coupling

map in Fig. 1c shows non-identical couplings between different pairs of qubits it might be a good sanity check to choose different qubits as the system qubits in successive experimental runs in order to support that variations in coupling strengths or gate infidelities do not impact the measurement outcome. ”

- **Our response:** We thank the Reviewer for the questions.

Figure 4: Ramsey experiments of Q_2 when Q_6 is in the state $|0\rangle$ and $|1\rangle$. Both qubits are located at the idle points. Circles and dashed lines represent experiment data and fitted data, respectively.

1. As we have mentioned above, the processor used in our experiments is the same as the one used in the Ref. 57 where $X/2$ gates of all qubits have been calibrated by the simultaneous random benchmarking and achieve high fidelities above 99% (see Supplementary Material of Ref. 57). In our experiments, we also calibrated X gates and $X/2$ gates of all qubits using quantum state tomography. We first apply a X ($X/2$) gate to the target qubit Q_j and apply I gates to other qubits, then apply tomography pulses to Q_j . The state fidelity between the ideal state $X|0\rangle$ ($X/2|0\rangle$) and the experimental state is denoted as $F_{X_{\text{single},j}}$ ($F_{X/2_{\text{single},j}}$). Furthermore, we also apply X ($X/2$) gates to all qubits simultaneously followed by tomography pulse of qubit Q_j to obtain the state fidelity $F_{X_{\text{all},j}}$ ($F_{X/2_{\text{all},j}}$). In Supplementary Note 1, we list state fidelities of all qubits in both situations. The medians of $F_{X_{\text{single},j}}$, $F_{X_{\text{all},j}}$, $F_{X/2_{\text{single},j}}$ and $F_{X/2_{\text{all},j}}$ are 99.87%, 99.32%, 99.95% and 99.76%, respectively. We also check the validity of our scheme to generate a random XY gate by applying a layer of random XY gates to all qubits simultaneously. The state fidelities of these random gates are shown in Fig. 2. The idle frequencies, fidelities $F_{X_{\text{single},j}}$, $F_{X/2_{\text{single},j}}$, $F_{X_{\text{all},j}}$ and $F_{X/2_{\text{all},j}}$ of all qubits are all given in the Tab. S1 of Supplementary Note 1.
2. The interactions between qubits are turned off during the single-qubit gates to

minimize the crosstalks between different qubit pairs. The frequencies of all qubits during single-qubit gates, which we call the idle points, are rearranged to cut off the interactions between any two qubits as shown in Fig. 3, where blue and red bars represent frequencies f_{10} and f_{21} of our transmon qubits. For any two qubits staying at their respective idle points, we perform two Ramsey experiments on one qubit, where the other qubit is in the ground state and the first excited state as shown in Fig. 4. The effect of crosstalks can be characterized by the Ramsey frequency difference δf . We have checked that all $|\delta f| \leq 0.1$ MHz. In addition, the high state fidelities $F_{X_{\text{all},j}}$ and $F_{X/2_{\text{all},j}}$ of all qubits also indicate that the interactions at idle points are deeply suppressed.

We add the discussion in the above paragraph about minimizing crosstalks between qubits during single-qubit gates in the Supplementary Note 1.

Figure 5: Negativity spectra and logarithmic negativities sampled from pseudo-random circuits for different groups of system qubits when $N = 9$. (a) Histograms for 10 choices of system qubits and ideal result. (b) Average logarithmic negativities for all choices of system qubits.

N	System qubits					
15	Q ₃	Q ₅	Q ₁₀	Q ₁₂	Q ₁₅	Q ₁₉
14	Q ₁	Q ₃	Q ₆	Q ₁₂	Q ₁₆	Q ₁₉
13	Q ₁	Q ₃	Q ₉	Q ₁₂	Q ₁₆	Q ₁₈
9	Q ₁	Q ₅	Q ₈	Q ₁₀	Q ₁₅	Q ₁₈
8	Q ₁	Q ₃	Q ₅	Q ₁₂	Q ₁₅	Q ₁₈
7	Q ₁	Q ₅	Q ₈	Q ₁₀	Q ₁₅	Q ₁₈

Figure 6: System qubits for different total qubit numbers N .

3. We also found that the variations of qubit-qubit coupling strengths do not impact the results. We substitute the free evolution operator generated by the non-uniform coupling strengths from experiments into the nine-qubit pseudo-random circuits

and consider all possible choices of system qubits with $C_9^6 = 84$ cases. In Fig. 5(a), we show the distribution of negativity spectra sampled from 20 instances where we choose 10 groups of system qubits shown in different histograms. They are all close to the ideal distribution sampled from random mixed states. In Fig. 5(b), we plot the average logarithmic negativities derived from the negativity spectra for 84 groups of system qubits, which are also very close to the ideal result. It means that the choice of system qubits does not impact the measurement outcome.

4. In our experiments, we choose the following groups of qubits as system qubits when total number of qubits decreases from 15 to 7 as shown in Fig. 6, which might be a sanity check as remarked by the Reviewer.

We add the discussion about the variations of qubit-qubit coupling strengths including Fig. 5 and Fig. 6 in the Supplementary Note 7.

- **Remark 6 of Reviewer #1:** “ In Fig. 2a the authors show the phase diagram for infinitely large N , and it is my understanding that it was derived in Ref. 22 for the large-system limit. Can the authors comment on the validity of this phase diagram for finite system (and environment) sizes, as used in their experiment? ”
- **Our response:** We thank the Reviewer for noting this important point. Indeed, there are a few differences between the phase diagram for the large-system limit and the phase diagram for finite system. There are two independent axes x, y in the phase diagram shown in Fig. 2a where $x = N_{A_1}/N$ and $y = N_A/N$. As shown in the main text, the more accurate transition point between PPT phase and NPT phase for finite system is

$$N_B = N_A + 2, \quad (7)$$

verified by the negativity spectrum illustrated in Fig. 2d-f. In the large-system limit where $N_A \rightarrow \infty$ and $N_B \rightarrow \infty$, we obtain the first phase boundary $y = 0.5$ shown as the yellow line in Fig. 2a. In our experiment, the corresponding phase transition occurs at $N_A = 6$ and $N_B = 8$. So the correct phase boundary in our system should be $y = 3/7 \approx 0.43$. The transition point between ME phase and ES phase is

$$|N_{A_1} - N_{A_2}| = N_B, \quad (8)$$

verified by the negativity spectrum illustrated in Fig. 2e-i. Eq. (8) can be rewritten as

$$y = \frac{1}{1 + |2x - 1|}, \quad 0 \leq x \leq 1, \quad 0 \leq y \leq 1, \quad (9)$$

which is the second phase boundary shown as the red line in Fig. 2a. In our experiment, the corresponding phase transition occurs at $N_{A_1}(N_{A_2}) = 2$, $N_{A_2}(N_{A_1}) = 4$ and $N_B = 2$, which corresponds to the point $(1/3(2/3), 3/4)$ and satisfies Eq. (9).

Therefore, we found the differences between the phase diagram for the large-system limit and the phase diagram for our system are small. For simplicity, we still use the

phase diagram for the large-system limit in Fig. 2a. In addition, we also mark the real phase boundaries in our experiments with dashed lines as shown in Fig. 3a and b.

- **Remark 7 of Reviewer #1:** “ The authors mention that they used five layers for their experiments in Fig. 2 and Fig. 3. However, in Fig. 4 they show that the experimentally extracted probability distributions are closest to the expected Porter-Thomas distribution for three layers. Can the authors please elaborate on this discrepancy? This suggests that the underlying probability distributions used for the negativity results do not follow a Porter-Thomas distribution. Logically, I would rather check that the probability distribution follows Porter-Thomas first (and find the optimal number of layers based on that) and then use this optimal layer number for the negativity experiments (basically show Fig. 4 first, followed by the negativity results). Can the authors comment why they did not adopt this strategy? ”

Figure 7: Distribution of probabilities of five-qubit bit-strings from three layer circuits, four-layer circuits, three-layer circuits with decoherence errors and four-layer circuits with decoherence errors $T_1 = 20 \mu s$ and $T_2 = 5 \mu s$. We sample 5000 circuit instances for each histogram.

- **Our response:** We thank the Reviewer for noting this point. As mentioned in the main text, we use four-layer circuits for $N = 7, 8$ and 9 and five-layer circuits for $N = 13, 14$ and 15 . In Fig. 4 of the main text, we illustrate the bit-string probability distributions for nine qubits, which we clarify in the Results Section and caption of the Fig. 4 in the revised manuscript. Although the output of three-layer circuits for nine qubits is closest to the Porter-Thomas (PT) distribution in experiments, we observe that three-layer circuits are not deep enough for the negativity spectra to converge (see Supplementary Note 3), suggesting the states to be not random enough, hence we implement deeper circuits in the negativity experiments.

The discrepancy that the optimal layer for observing nine-qubit PT distribution seems to be three can be explained by the decoherence errors in the experiments. We simulate the distribution of probabilities of output bit-strings with decoherence errors by using Lindblad master equation to capture the effect of finite energy relaxation time T_1 and Ramsey dephasing time T_2 of qubits. In Fig. 7, the distribution sampled from three-layer circuits of five qubits with $T_1 = 20 \mu\text{s}$ and $T_2 = 5 \mu\text{s}$ is closer to the distribution sampled from four-layer circuits without decoherence errors, compared with the three-layer circuits without decoherence errors. The KL divergences D_{KL} between the PT distribution and the output of three-layer circuits, four-layer circuits, three-layer circuits with decoherence errors and four-layer circuits with decoherence errors are 0.0092, 0.0017, 0.0032 and 0.0104, respectively. Therefore, if there are no decoherence errors, the output of four-layer circuits is closer to the PT distribution. The decoherence errors seem to displace the distributions such that the output of three-layer circuits approaches to the PT distribution.

We add Fig. 7 and the discussion in this response about the impact of decoherence errors on distributions of output bit-strings in the Supplementary Note 9. We also add a sentence “Although the output of three-layer circuits for nine qubits is closest to the PT distribution in experiments, we observe that three-layer circuits are not deep enough for the negativity spectra to converge (see Supplementary Note 3), suggesting the states to be not random enough, hence we implement deeper circuits in the negativity experiments. See also Supplementary Note 9 where we discuss the effect of decoherence on bitstring probability distributions.” at the end of the last paragraph of the Results Section.

- **Remark 8 of Reviewer #1:** “ The authors do not mention the anharmonicity of their qubits nor the effect of the second excited levels of their transmon qubits. Did the authors check the higher-level excitations or can they motivate that there is no adverse effect for the probability distribution measurements, density matrix extraction, or their experimental scheme in general? Since the experimental realization of quantum state tomography in a six-qubit system is rather expensive in terms of number of measurements I would appreciate if the authors could provide some details on their experimental methods. I was not able to find the Supplementary Notes, where I assume the authors have made more explanations on the experimental details, but in any case I would appreciate it if the authors could add a few additional explanatory sentences on their experimental procedure in the main text or in the Methods section. This also applies to their correction scheme of data points, which deserves at least a short explanation in the main text. ”
- **Our response:**
 1. We thank the Reviewer for raising this important point. The anharmonicities of our qubits η_j all lie within the range $-260 \text{ MHz} < \eta_j/2\pi < -240 \text{ MHz}$. We insert the information about anharmonicity into the description of experimental device

in the Methods Section of the main text. In experiments, all single-qubit pulses are modulated by DRAG (PRL 103, 110501 (2009)) to remove most of leakage during single-qubit gates, verified by the high-fidelities of single-qubit gates. Since all coupling strengths J_{ij} between qubits are far less than η_j , i.e., $J_{ij}/|\eta_j| < 1/100$, we think that the effect of the second excited levels during free evolution is minor. To quantitatively analyze the relevant effect, we numerically calculate the growth of leakage $P_j \equiv |\langle 2_j | \rho_j(l) | 2_j \rangle|^2$ over the circuit layer by substituting the fully connected Bose-Hubbard model into the 40-ns free evolution part of random circuits, where $|2_j\rangle$ denotes the second excited state of Q_j . $\rho_j(l)$ is the reduced density matrix of Q_j after a l -layer circuit. For simplicity, we set $J_{ij}/2\pi = 2$ MHz for $i, j = 1, 2, \dots, N$ such that $P \equiv \overline{P_j}$ for $j = 1, 2, \dots, N$ where “—” indicates the statistical average of different circuit instances. As shown in Fig. 8(a) and (b), the average leakage increases over the layer. Furthermore, the average leakage also increases with the qubit number N linearly when the circuit layer is fixed, which can be explained by the fact that the leakage rate of one qubit is proportional to the number of its coupling qubits. As a comparison, we also calculate the leakage of a one dimensional Bose-Hubbard model with a periodic boundary condition as shown in Fig. 8(c) and (d). It is clear that the leakage still increases over the layer but is almost independent with the qubit number N , since each qubit is always coupled with two adjacent qubits.

We sample 200 circuit instances for $6 \leq N \leq 8$ and 50 instances for $9 \leq N \leq 12$ in Fig. 8. To estimate the leakage for larger systems, we fit the leakage over the qubit number by linear functions and extrapolate it to $N = 13, 14$ and 15 . For the 15-qubit fully connected system, the leakage error after 5 layers is about 3×10^{-3} which is close to a single-qubit gate error. Therefore, we think that the effect of the second excited levels in our system is not adverse.

We add the discussion in this response about the effect of the second excited levels of transmon qubits including Fig. 8 in the Supplementary Note 8.

2. We thank the Reviewer for the suggestion. We confirm that we have submitted a Supplementary Material named “supplementary.pdf”. We also add a short introduction to our QST procedure in the second paragraph of the Results Section of the main text. We also add a short explanation to the correction scheme of data points in Fig. 3 in the fifth paragraph of the Results Section where we discuss the results in Fig. 3.
- **Remark 9 of Reviewer #1:** “ In Fig. 3a, why is the negativity for $N_B = 7$ so small, when according to the text we would expect non-zero negativity in the NPT region? Might a logarithmic scaling of the vertical axis in Fig. 3a help to better visualize these (small) values? ”
 - **Our response:** We thank the Reviewer for the question. The negativity for $N_B = 7$ is small which can be intuitively explained by $N_A = 6 < N_B$. When system qubits is

less than environment qubits, most of them are still entangled with environment qubits such that the entanglement between themselves is small. The ideal average logarithmic negativity for $N_B = 7$ is 0.04, which can be obtained from the semi-circle law, i.e.,

$$P(\xi) = \frac{2L_A}{\pi a^2} \sqrt{a^2 - \left(\xi - \frac{1}{L_A}\right)^2}, \quad \left|\xi - \frac{1}{L_A}\right| < a. \quad (10)$$

Since the negativities for $N_B = 9$ and 8 are both zero, it might not be a good way to show these values with a logarithmic scale in the vertical axis. Instead, we add a subfigure in Fig. 3a which zooms in the small negativities from $N_B = 9$ to $N_B = 7$ to help visualize.

- **Remark 10 of Reviewer #1:** “ The authors explain how they divide their circuit into three parts A_1 , A_2 , and B in the last paragraph of page 3, but already use this notation in the context of negativity. ”
- **Our response:** We thank the Reviewer for noting this point. We explain how we divide our circuit into three parts A_1 , A_2 and B before using this notation to derive Eq. (2) in the second paragraph of the Results section of the revised manuscript.
- **Remark 11 of Reviewer #1:** “ Fig 1a is missing a scale bar. In Fig. 1c the authors should invert their color bar, such that highest coupling strengths are not white. ”
- **Our response:** We thank the Reviewer for these valuable suggestions. We add a scale bar at the lower right corner of Fig. 1a. We also adjust the color bar of Fig. 1c such that the highest coupling strengths are more conspicuous than smallest coupling strengths.

Figure 8: The leakage over the layer sampled from pseudo-random circuits. We sample 200 instances for $6 \leq N \leq 8$ and 50 instances for $9 \leq N \leq 12$. The leakages are fitted by linear functions. The free evolution in (a) and (b) is governed by a fully connected Bose-Hubbard model while in (c) and (d) is governed by a one dimensional Bose-Hubbard with a periodic boundary condition.

Responses to the Reviewer #2

- **Remark 1 of Reviewer #2:** “ The work titled Observation of entanglement transition of pseudo-random mixed states by Tong Liu and coworkers have studied the Haar measure random states generated using superconducting processor. They have demonstrated the entanglement transition as predicted by works published in Refs. [19,20,21,22]. This is a very important experimental demonstration of its kind. The experimental results agrees very well the theoretical once considering the error introducing factors like decoherence. This system will also act as a testbed for testing other related results. The paper is well written with all key results explained very well. The figures are comprehensible. Key papers are included in the references. ”
- **Our response:** We would like to express our sincere gratitude to the Reviewer for reviewing our paper and kind words such as “This is a very important experimental demonstration of its kind. The experimental results agrees very well the theoretical once considering the error introducing factors like decoherence”. We also thank the Reviewer for their valuable suggestions and comments.
- **Remark 2 of Reviewer #2:** “ The authors should specify the total number of realizations i.e. total number of states generated (experiments and simulations) to get the plots of figures 2, 3 and 4. Also the time taken to generated one full state and its QST. ”
- **Our response:** We thank the Reviewer for the suggestion. The total numbers of states generated for Figures 2 and 3 are both 20 which are given in the fourth and fifth paragraphs of the Results Section. The total number of states generated in Figure 4 is 300 which we add in the last paragraph of the Results Section. The time taken to generate one full state is dependent on the layer of the circuit. Each layer of our circuit is composed of two layers of single-qubit gates and one layer of global entangling gate. The time taken is 15 ns to generate a single-qubit gate and 40 ns to generate a global entangling gate. We specify them in the first paragraph of the Results Section. The whole pulse sequence for the QST including the state generation, the tomography operation and the readout takes about 2 μ s. We add the time taken about QST in the second paragraph of Results Section.
- **Remark 3 of Reviewer #2:** “ Some important references on QST can be added since it is one of the main tool in this work. ”
- **Our response:** We thank the Reviewer for the helpful suggestion. We have added the following references on QST in the third paragraph of Introduction Section where we firstly mention the QST.
 - D. F. V. James, P. G. Kwiat, W. J. Munro, and A. G. White, Measurement of qubits, Phys. Rev. A 64, 052312 (2001).
 - D. Gross, Y.-K. Liu, S. T. Flammia, S. Becker, and J. Eisert, Quantum state

- tomography via compressed sensing, *Phys. Rev. Lett.* 105, 150401 (2010).
- M. Cramer, M. B. Plenio, S. T. Flammia, R. Somma, D. Gross, S. D. Bartlett, O. Landon-Cardinal, D. Poulin, and Y.-K. Liu, Efficient quantum state tomography, *Nat. Commun.* 1, 149 (2010).
 - G. Torlai, G. Mazzola, J. Carrasquilla, M. Troyer, R. Melko, and G. Carleo, Neural-network quantum state tomography, *Nat. Phys.* 14, 447 (2018).
 - A. Rocchetto, S. Aaronson, S. Severini, G. Carvacho, D. Poderini, I. Agresti, M. Bentivegna, and F. Sciarrino, Experimental learning of quantum states, *Sci. Adv.* 5, eaau1946 (2019).
 - H.-Y. Huang, Learning quantum states from their classical shadows, *Nat. Rev. Phys.* 4, 81 (2022).
- **Remark 4 of Reviewer #2:** “ Authors can add the following relevant reference in the introduction: Experimental Realization of a Measurement-Induced Entanglement Phase Transition on a Superconducting Quantum Processor by Jin Ming Koh, Shi-Ning Sun, Mario Motta, and Austin J. Minnich (<https://arxiv.org/abs/2203.04338>). ”
 - **Our response:** We thank the Reviewer for the helpful suggestion. We have added the relevant reference in the second paragraph of the Introduction Section.
 - **Remark 5 of Reviewer #2:** “ Finally, I feel that the authors have experimentally demonstrated an important theoretical result on the entanglement transition. This work deserves the publication in Nature Communications and I am happy to recommend this work for publication. ”
 - **Our response:** We appreciate the Reviewer’s very positive evaluation of our work. We sincerely thank the Reviewer for recommending our work for publication in Nature Communications.

Responses to the Reviewer #3

- **Remark 1 of Reviewer #3:** “ This work is a very nice experimental study of transitions in the structure of entanglement in quantum states generated by pseudorandom circuits. The novel experimental aspect of this study is the use of nonlocal connectivity, which allows for rapid scrambling. The reduced density matrices are characterized by standard tomographic techniques (and hence so is the negativity). The bit-string probabilities are also computed and found to agree with expectations for random circuits.

This work performs a nontrivial experimental test of an interesting theory prediction and as such I recommend its publication. The paper is clearly written and I have no substantive criticisms. ”

- **Our response:** We would like to express our sincere gratitude to the Reviewer for reviewing our paper and a very positive evaluation of our work. We also sincerely thank the Reviewer for recommending our work for publication.

REVIEWERS' COMMENTS

Reviewer #1 (Remarks to the Author):

The authors have addressed all remarks from my previous report satisfactorily. I would like to thank the authors for their comprehensive responses and for providing additional measurements that further substantiate and support the presented results.

I recommend the manuscript for publication in Nature Communications.